# Lipid Biomarkers and Atherosclerosis—Old and New in Cardiovascular Risk in Childhood

**DOI:** 10.3390/ijms24032237

**Published:** 2023-01-23

**Authors:** Mirjam Močnik, Nataša Marčun Varda

**Affiliations:** 1Department of Paediatrics, University Medical Centre Maribor, Ljubljanska Cesta 2, 2000 Maribor, Slovenia; 2Faculty of Medicine, University of Maribor, Taborska 8, 2000 Maribor, Slovenia

**Keywords:** lipids, children, cardiovascular risk, biomarkers

## Abstract

Lipids are a complex group of molecules in the body, essential as structural, functional and metabolic components. When disbalanced, they are regarded as a cardiovascular risk factor, traditionally in cholesterol level evaluation. However, due to their complex nature, much research is still needed for a comprehensive understanding of their role in atherosclerosis, especially in the young. Several new lipid biomarkers are emerging, some already researched to a point, such as lipoproteins and apolipoproteins. Other lipid molecules are also being increasingly researched, including oxidized forms due to oxidative inflammation in atherosclerosis, and sphingolipids. For many, even those less new, the atherogenic potential is not clear and no clinical recommendations are in place to aid the clinician in using them in everyday clinical practice. Moreover, lipids’ involvement in atherogenesis in children has yet to be elucidated. This review summarizes the current knowledge on lipids as biomarkers of cardiovascular risk in the paediatric population.

## 1. Introduction

Cardiovascular diseases (CVD) are still the leading cause of morbidity and mortality worldwide [1]. The underlying pathological mechanism is atherosclerosis—a narrowing of the arteries caused by a buildup of plaque, leading to insufficient blood flow to vital organs, mainly the heart and brain [2]. More than 100 years ago, Virchow discovered that the plaque, called atheroma, contained a yellow fatty substance (later identified as cholesterol) and already suggested the role of lipids in atherosclerosis [3]. Afterwards, several lipid classes have traditionally been associated with CVD [4], but many patients have traditional lipid levels within the recommended range, presenting a need for the discovery of additional lipid biomarkers [5].

Atherosclerosis has already begun in childhood [6]. Early atherosclerotic changes found in autopsied infants and adolescents [6,7] who died from non-CVD demonstrated an association between atherosclerosis in the coronary arteries and abdominal aorta with classic cardiovascular risk factors, such as dyslipidemia, hypertension, impaired glucose tolerance and obesity [6,7]. Additionally, studies are showing earlier occurrence of CVD in adults who had cardiovascular risk factors present as children [8,9], indicating the need for cardiovascular assessments for young people, including a comprehensive lipid status evaluation.

Lipids are hydrophobic molecules playing key roles in cellular membranes, cell signaling and cell metabolism [10]. They can be classified in eight lipid classes: fatty acids, glycerolipids, glycerophospholipids, polyketides, prenol lipids, saccharolipids, sphingolipids and sterol lipids. Sequencing of the human genome has allowed further research, and lately attention has shifted to postgenomic technologies, such as metabolomics [5]. Metabolomics is defined as the comprehensive analysis of metabolites in a biological specimen; current metabolomic technologies are capable of precise analyses of hundreds to thousands of metabolites [11]. However, lipids present a demanding challenge for comprehensive analysis due to their complex composition and the high number of different lipid specimens and their modifications. This is reflected also by the fact that lipidomics has not been as widely used and published in recent years as other -omic (e.g., proteomics, genomics) technologies [5]. For the same reason, cardiovascular research has focused mostly on the role of lipid classes, such as total triglycerides, rather than molecules themselves [5].

This review highlights the role of lipids in cardiovascular risk assessment in children. First, the systematic search approach, with comparison of previous similar review papers, is described. Next, the relevant lipids in cardiovascular risk assessment in children have been divided into three groups according to their evidence-based meaning in clinical practice. For each, the physiological function, as far as it is known, has been described along with current evidence in the context of cardiovascular risk in the paediatric population. We conclude by emphasizing the need for further research of lipid biomarkers in cardiovascular risk assessment in children.

## 2. Methods and Previous Studies

We searched for studies reporting on lipid-based metabolites (e.g., lipid biomarkers) and their association with cardiovascular risk in children. Several layers of keyword search criteria were used in the PubMed, Web of Science and Google Scholar databases in the years from 2000 to 2022: children, paediatric, lipids, biomarkers, cardiovascular risk. The results were filtered according to their significance and association with the paediatric population. None of the reviews found presented lipid biomarkers as cardiovascular risk factors in the paediatric population, which makes our review unique. Therefore, reviews pertaining to adults were analyzed in association with a manual search of the relevant mentioned biomarkers in children. The most important good-quality studies are presented in Table 1 with comparison to our review.

Countless other lipid biomarkers are being researched; however, only the most clinically relevant, promising and established in children were included. For the purpose of this review, lipid biomarkers have subsequently been divided into traditional, newer and research biomarkers, as shown in Table 2. All of them were also researched in the adult population. The traditional ones are regarded as commonly accepted cardiovascular risk biomarkers, and the newer ones are associated with the traditional one and are being increasingly used in clinical practice; however, their role has not yet been established in full. The last group of lipids presents biomarkers that are being increasingly investigated in children due to their potential as cardiovascular risk biomarkers, but they have not yet been incorporated into clinical practice, nor has their role in cardiovascular risk been established. 

## 3. Lipid Determination Methods

The most common lipid determination methods include nuclear magnetic resonance and tandem mass spectrometry. The first provides a method for distinguishing and quantifying a wider range of lipoprotein subclassess. On exposure to a magnetic field, distinct lipoprotein subclasses emit a unique signal that is directly proportional to their concentration [5]. It provides a useful tool for lipid structure analysis in both solution and solid states; however, it lacks sensitivity and requires more samples when compared with mass spectrometry [5]. The latter is the method of choice for the identification and quantification of lipid species with high sensitivity and selectivity. Before detection, lipid species should be separated to suppress the ionization of coeluting low-abundant compounds; this is most commonly achieved using liquid chromatography. In lipidomics, bottom-up and top-down approaches are used. In the bottom-up approach, lipids are analyzed according to their specific fragmentation pattern obtained by tandem mass spectrometry; in the top-down approach, intact lipid species are identified and quantified with lipid discrimination by their exact mass. Quantification can be either relative or absolute [5].

## 4. Traditional Lipid Biomarkers

Total cholesterol is the most known cardiovascular risk factor among lipids. The molecular structure of cholesterol is lipoprotein, and it has a central hydrophobic core (primarily cholesterol esters and triglycerides) surrounded by a hydrophilic membrane (phospholipids, free cholesterol and apolipoproteins), as presented schematically in Figure 1. 

Lipoproteins belong to the group of sterol lipids, with the exception of triglycerides, which are fatty acids. Measuring total cholesterol includes measuring chylomicrons, chylomicron remnants, very low-density lipoproteins (VLDL), intermediate-density lipoproteins (IDL), low-density lipoproteins (LDL), high-density lipoproteins (HDL) and lipoprotein (a) (Lp(a)). Triglycerides are major lipids located in chylomicrons (and their remnants), VLDL and IDL; meanwhile, cholesterols are abundant in LDLand HDL, but are also present in IDL and chylomicron remnants [13]. 

Chylomicrons are made in the intestine by the dietary lipids, and their size depends on the amount of fat ingested. They are rich in triglycerides and are transported to the liver and peripheral tissues. Chylomicron remnants are the result of the removal of triglycerides from chylomicrons in the peripheral tissues. They have more cholesterol and are pro-atherogenic. VLDL are produced by the liver; meanwhile, the IDL are the result of the removal of triglycerides from VLDL by muscle and adipose tissue. They are pro-atherogenic. LDL are derived from IDL and VLDL and contain even more cholesterol, carrying the majority of the cholesterol in the circulation. HDL perform the reverse cholesterol transport from peripheral tissues to the liver and are regarded as anti-atherogenic [13]. The relationship between dietary fat, basic lipid metabolism and atherogenesis is presented in Figure 2.

### Hypercholesterolemia and Cardiovascular Risk in Children

Hypercholesterolemia is one of many cardiovascular risk factors in childhood, and it is indisputable that it can be identified from early years on as a prediction of future cardiovascular risk [14]. It is often associated with other risk factors, especially obesity, which also carries an increased risk of hypertension, insulin resistance and overall atherosclerotic CVD development [15]. The risk of hypercholesterolemia development in obesity is increased with central obesity and increased waist circumference [16]. In children with obesity, almost half showed an abnormal lipid profile with increased LDL and triglycerides and decreased HDL values, with the latter two being associated with an increased risk of metabolic syndrome development [17,18].

Total cholesterol and LDL in children with obesity have been associated with fatty deposits in the aorta and coronary arteries with an onset in childhood [19]; additionally, in familial hypercholesterolemia, a genetic disorder of lipid metabolism not so rare in the paediatric population, an increase in carotid intima media thickness (cIMT) was found independently from plasma lipid levels and measures of adiposity [20,21].

Although CVD seem to be rare in childhood, the atherosclerotic process begins in childhood. Early and prolonged exposure to cardiovascular disturbance, such as dyslipidemia, contributes to premature CVD in ever-yonger adult populations [22]. Paediatric dyslipidemia was associated with increased cIMT and therefore with cardiovascular risk in young adults [23,24]. LDL and BMI in childhood were the best predictors of adverse levels of cholesterol and triglycerides in young adulthood [24,25]. Different ratios between traditional markers have been developed aiming to better evaluate cardiovascular risks, such as atherogenic index of plasma (logarithmically transformed ratio of molar concentrations of triglycerides to HDL) [26] or triglyceride/HDL ratio [27]; results show that they could be useful in identifying children at risk of dyslipidemia and metabolic syndrome [27].

Obesity in childhood presents an increased risk for many cardiovascular risk factors in young adulthood, among which dyslipidemia plays an important role. However, the risks of adverse outcomes among overweight or obese children who became nonobese by adulthood were similar to those among individuals who were never obese [28], emphasizing the role of early intervention at a young age. Lifestyle change has also been associated with an improved lipid profile, especially with decreased triglycerides [29,30].

## 5. Newer Lipid Biomarkers

In this group we included Lp(a) and apolipoproteins, which play a crucial role in lipoprotein metabolism. They include apolipoprotein A-I (APO-AI), apolipoprotein A-II (APO-AII), apolipoprotein A-IV (APO-AIV), apolipoprotein A-V (APO-AV), apolipoprotein B-48 (ABO-B48), apolipoprotein B-100 (APO-B100), apolipoproteins C (APO-C), apolipoproteins E (APO-E) and apolipoprotein(a) (apo(a)) [13]. Not all seem to be clinically relevant. Their role in cardiovascular assessment is controversial—already more than forty years ago their utility was shown in atherosclerosis determination [31]; later, however, some studies showed no advantages over conventional lipoproteins in predicting the risk of CVD in the adult population [32,33].

### 5.1. Lipoprotein(a) and Apolipoprotein(a)

Lp(a) is a lipoprotein that is very similar to LDL but that has a unique apolipoprotein—apo(a). Its genetic sequence is highly variable, resulting in a protein with molecular weight ranging from 300 to 800 kDa [34]. Apo(a) inhibits plasminogen activation [13]; however, the physiological function of Lp(a) is unknown [13]. Several analyses have demonstrated an association between Lp(a) and coronary heart disease, especially in particles with smaller size apo(a) isoforms [34]. In adult patients with elevated Lp(a), a high prevalence of atherosclerotic cardiovascular disease, familial hypercholesterolemia and a familial history of premature atherosclerotic cardiovascular disease was noted [35,36]. Similarly, meta-analyses showed an increment of Lp(a) in children with a family history of CVD and a positive association between Lp(a) and total cholesterol, LDL and apolipoprotein B levels [37]. Even more worryingly, Lp(a) levels were highest in children with a family history of premature CVD, increasing the need for Lp(a) screening in children with an increased risk of premature CVD and cardiovascular events [38,39]. Moreover, it has been shown that cardiovascular and cerebrovascular events could be predicted by an increased Lp(a) level independently of other cardiovascular risk factors [40]. A positive association between childhood arterial ischemic stroke and Lp(a) was found; however, this is a very rare event in childhood and the data is scarce [41]. According to Goldenberg et al., elevated Lp(a) and apo(a) could contribute to atherogenesis and ischemic events by impairing fibrinolysis [42].

Anyway, subclinical atherosclerosis seems less problematic in children with higher Lp(a) levels, demonstrated by their similar cIMT and pulse wave velocity (PWV), another marker of subclinical atherosclerosis [43], to children with normal Lp(a) levels. Recommendations for young individuals on the use of Lp(a) in clinical practice are limited and comprehensive guidelines are still needed to facilitate clinical decision making for the screening and management of elevated Lp(a) in the paediatric population [44].

### 5.2. Apolipoproteins

Apolipoproteins are a part of lipoproteins present in the hydrophilic membrane surrounding the hydrophobic core [13], as already presented schematically in Figure 1.

They have a structural role and also act as ligands for lipoprotein receptors, guiding the formation of lipoproteins and serving as activators or inhibitors of enzymes involved in the metabolism of lipoproteins [13].

#### 5.2.1. Apolipoproteins A

Apolipoproteins A are a diverse group of apolipoproteins with different functions. Four molecules are discussed in the literature (APO-AI, -AII, -AIV and -AV) [13].

APO-AI is a major structural protein of HDL, accounting for more than two thirds of HDL protein. It is also an activator of lecithin cholesterol acyltransferase (LCAT), which converts free cholesterol to cholesteryl ester. High levels of APO-AI are considered anti-atherogenic [13]. Inversely, low APO-AI levels were associated with endothelial dysfunction in obese children [45]. Interestingly, there is also growing evidence that APO-AI’s synthesis is regulated with vitamin D levels, with a positive link between increasing vitamin D levels and plasma APO-AI along with HDL reported [46].

APO-AI is the most abundantly researched apolipoprotein A, mostly in a ratio with apolipoproteins B, which was shown to have a better predictive role in cardiovascular risk assessment in adults [47], with elevated levels of APOB and reduced levels of APO-AI associated with increased cardiac events [48]. Similar imbalances were seen in younger people, especially in relation to obesity and metabolic syndrome in the paediatric population [49,50]. APOB–APO-AI ratio was significantly associated with visceral obesity in male children and adolescents [51]. Both APOB and APOB–APO-AI ratio were directly associated with pulse wave velocity, a measure of subclinical atherosclerosis [52]. Childhood values of both also predicted cIMT and brachial endothelial function in adulthood [53], emphasizing the role of preventive screening and management in the paediatric population. Both APO-AI and APOB in children with obesity were also associated with other cardiovascular risk factors, such as elevated blood pressure, glucose metabolism and inflammation [54,55]. Even after adjustment for childhood BMI, systolic blood pressure and other lipoprotein measures, APOB and APOB–APO-AI ratio remained significant predictors of cardiovascular risk; however, the significance of APO-AI decreased to non-significant levels [56]. Interestingly, inflammation itself was associated with an unfavorable lipid profile in children with/after Kawasaki disease. Some of the participants had already had coronary aneurysms and were associated with lower HDL and APO-AI and increased APOB levels and PWV. The latter two were also higher in children after Kawasaki disease without coronary damage present [57,58].

APO-AII is the second most abundant protein in HDL protein. Its role is unclear but it seems to be a strong predictor for CVD [13]. Higher levels of APO-AII were associated with increased arterial stiffness in children and adolescents, with and without adjusting for fat mass present [59].

APO-AIV is associated with HDL, chylomicrons and lipoprotein-free fractions. Its precise physiological role is not yet determined; however, studies indicate that it might be involved in the regulation of food intake [13]. APO-AV activates lipoprotein lipase (LPL) and therefore plays a role in the metabolism of triglyceride-rich lipoproteins [13]. APO-AV genetic variants have been associated with triglyceride levels related to sex, and also with fat-soluble vitamin E, which has antioxidant activity [60].

#### 5.2.2. Apolipoproteins B

Apolipoprotein B is a product of a single apolipoprotein B gene expressed in the liver and intestine. It is later modified with mRNA editing, resulting in APO-B48 and APO-B100. APO-B48 is a major structural protein in chylomicrons and chylomicron remnants; meanwhile, APO-B100 is a major structural protein in VLDL, IDL and LDL. It is also a ligand for LDL receptors and thereby plays an important role in clearing lipoprotein particles. Elevated levels of APO-B100 are associated with a pro-atherogenic state [13]. Underdiagnosed state, familial defective apolipoprotein B-100 and an autosomal dominant genetic disorder of lipid metabolism are also associated with hyperlipidemia and an elevated risk of atherosclerosis; however, the elevation of LDL is usually milder than that of familial hypercholesterolemia (caused by a mutation in the receptor for LDL). Population studies suggest that the mutation is present in 0.1% of Northern Europeans and US Caucasians [61].

More commonly, the ratio between apolipoproteins B and APO-AI is researched, as has already been discussed above, emphasizing the proatherogenic effect of high APOB values, usually associated with obesity and subclinical atherosclerosis [49,51,52]. Additionally, it has more commonly been associated with metabolic syndrome and cardiovascular events [62,63]. Increased APOB levels have also been seen among healthy adolescents frequently exposed to tobacco smoke, along with the arterial changes of subclinical atherosclerosis [64].

As other apolipoproteins present a possible therapeutic target, so does APOB, and a clinical trial with mipomersen, an antisense oligonucleotide inhibitor of APOB, was conducted. A double-blind, placebo-controlled study showed significant reductions of LDL, APOB and Lp(a) in adult patients with hypercholesterolemia at risk of coronary heart disease not controlled by existing therapies [65].

#### 5.2.3. Apolipoproteins C

APO-C lipoproteins are associated with chylomicrons, VLDL and HDL. APO-CI is an LCAT activator. APO-CII is a cofactor for LPL and therefore enhances triglyceride hydrolysis. Loss of its function results in marked hypertriglyceridaemia. APO-CIII is on the other hand an inhibitor of LPL, and it also inhibits the ligation of triglyceride-rich lipoproteins with their receptors. Loss of its function leads to hypotriglyceridemia and a reduced risk of CVD. Furthermore, inhibition of APO-CIII leads to a decrease in serum triglycerides, also observed in patients with LPL deficiency indicating another mechanism of triglyceride modulation independently of the regulation of LPL [13]. According to one study on adults, APO-CIII levels were significantly increased in patients with metabolic syndrome, with a further elevation in patients who also had coronary artery disease. A specific genetic variant was also associated with increased triglycerides, APO-CIII and coronary artery disease in the same cohort of patients. Interestingly, in patients with a specific genetic variant, obesity was less frequent than in patients without the variant but with metabolic syndrome present [66]. Both APO-CII and -CIII have been associated with increased arterial stiffness in children with and without adjusting for fat mass present [59]. Both were also researched in children with chronic kidney disease, and both correlated with total cholesterol, triglycerides and LDL. APO-CII was associated with high left ventricular mass and abnormal ambulatory blood pressure monitoring, which was also associated with APO-CIII [67].

#### 5.2.4. Apolipoproteins E

APO-E acts as an exchanger between lipoprotein particles and is associated with chylomicrons, chylomicrons remnants, VLDL, IDL and HDL. There are three common genetic variants: APO-E2, APO-E3 and APO-E4. APO-E3 and APO-E4 are ligands for LDL receptor, while APO-E2 only poorly binds to it; patients who are homozygous for this variant can develop familial dysbetalipoproteinemia. APO-E4 has also been associated with an increased risk of both atherosclerosis and Alzheimer’s disease [13]. The latter is attributable to its function in the nervous system to distribute lipids amongst cells [68]. Generally, APO-E2 is most favorable, and APO-E4 least favorable, for cardiovascular in neurological health [68]. One study observed a similar trend in the paediatric population [69], with differences in clustering cardiovascular risk factors according to the APO-E phenotype already formed after birth due to the influence of genetic and environmental factors on serum cholesterol levels, BMI, blood pressure and insulin resistance [69,70]. Later, it was shown that APO-E’s impact on glucose metabolism is less pronounced and does not seem to be associated with APO-E polymorphism [71]. A follow-up from childhood to adulthood showed that APO-E polymorphism is associated with lipid levels, especially LDL, at different ages, with APO-E4 polymorphism associated with consistently higher values of LDL and lower values of HDL and APO-AI. Similarly, in the same study APO-E2 polymorphism was associated with lower LDL [72]. More troublesomely, APO-E4 was also associated with a family history of premature coronary artery disease [73]. Moreover, the desired responses to physical and dietary interventions were least successful in the APO-E4 phenotype variant [74,75].

As in the adult population, the genetic variant of APO-E in the setting of nervous function and brain injury has also been questioned and is also associated with the function of cell maintenance and repair [76]. However, so far studies have been inconclusive, with some showing no meaningful effects on cognitive performance in the paediatric population [77] and others showing an effect on neurobehavioral performance, particularly spatial memory, in school-aged children [78].

## 6. Research Lipid Biomarkers

In this group, we present biomarkers differing from traditional cholesterols and triglycerides as well as from their attached apolipoproteins that could present novel biomarkers for cardiovascular risk in children or even a therapeutic target in the future.

### 6.1. Oxidized Lipid Molecules

Oxidized lipid molecules have been modified by lipid peroxidation, with the modification of both lipid and protein components promoting inflammatory and immunological mechanisms that lead to the formation of macrophage foam cells. The concept that the oxidation of lipoproteins is central in the pathogenesis of atherosclerosis was reported decades ago, and new reports of the relationship between oxidized lipids and subclinical atherosclerosis have since been incoming in the form of both in vitro and in vivo studies [79]. In favour of oxidizing inflammatory processes, the plasma levels of myeloperoxidase, an enzyme that generates the strong oxidizing agent hypochlorous acid, have been associated with the risk of myocardial infarction and endothelial dysfunction [80].

The most commonly researched oxidized lipoprotein is oxidized LDL (Ox-LDL), which has already been associated with subclinical atherosclerosis, coronary and peripheral arterial disease, acute coronary syndrome and vulnerable atherosclerotic plaques in human studies in the adult population [81]. It has also been associated with chronic conditions, promoting CVD, diabetes mellitus, obesity and metabolic syndrome [82]. Ox-LDL increases significantly across BMI groups in the paediatric population, suggesting a higher level of oxidative stress and inflammation in childhood (extreme) obesity [83]. The correlations are more pronounced in association with percent body fat, waist circumference, abdominal visceral fat and abdominal subcutaneous fat [84]. In a study of subclinical atherosclerosis, Ox-LDL was associated with estimates of stiffness and thickness; however, it was not seen as different from traditional risk factors (HDL, triglycerides) [85]. Reduced endothelial function with impaired nitrate-mediated dilatation and increased Ox-LDL has also been shown in the paediatric population, suggesting that primary nitrate tolerance occurs in children at risk for atherosclerosis [86]. However, not all studies have demonstrated an elevation of Ox-LDL in the paediatric population. For instance, in a study of adolescents with polycystic ovary syndrome Ox-LDL was not found to differ according to BMI status, and was also not found to be different from healthy lean controls [87].

Other factors might contribute to Ox-LDL levels, such as bilirubin levels, which can prevent lipid oxidation in vitro; it has been shown that lower levels of bilirubin contribute to increased Ox-LDL in obese children and adolescents [88]. Similarly, levels of Ox-LDL have been associated with carbon load and the effect of traffic air pollution [89]. Lifestyle modifications, physical activity, zinc supplementation and dietary changes also contribute to lower Ox-LDL [90,91,92,93].

Other lipoproteins are being oxidized in the process; however, research on the paediatric population is scarce. With higher myeloperoxidase levels, higher oxidized HDL (Ox-HDL) was observed, reflecting higher oxidative stress in children [94]. To some extent, oxidized phospholipids (Ox-PL) in APO-B100 lipoproteins have been researched, albeit mainly in the adult population, showing elevated levels of Ox-PL/APOB in the presence and progression of coronary, femoral and carotid artery diseases, with an additional increment with acute coronary syndromes and the prediction of adverse events (myocardial infarction, stroke, need for revascularization, death) [95]. The predictive value of Ox-PL/APOB is amplified by Lp(a) and phospholipases [95,96]. Additionally, Lp(a) has been shown to be the carrier of oxidized phospholipids, suggesting an additional mechanism of its pro-atherogenic effect in the general population [80,97]. A population-based study demonstrated that Ox-PL/APOB levels predict 10-year CVD event rates independently of traditional risk factors and Framingham Risk Score [98], indicating the need for research on novel biomarkers of lipid peroxidation in the paediatric population.

Lipid peroxidation byproducts, such as thiobarbituric acid reactive substances (TBARS), malondialdehyde (MDA) and 8-isoprostane F_2ɑ_, have also been associated with severe obesity in children [99] and could serve as an additional potential cardiovascular biomarker.

### 6.2. Lipoprotein X

Lipoprotein X (Lp-X) is an abnormal cholesterol-rich lipoprotein particle that can be found in familial lecithin:cholesterol acyltransferase deficiency, a rare genetic disease, and several hepatic disorders (primary biliary cirrhosis, primary sclerosing cholangitis, cholestatic liver disease, chronic alcoholism) with cholestatic liver disease [100]. Cases with severe and extreme hypercholesterolemia have been reported in the paediatric population, mediated by lipoprotein X, in rare cases of intra- and extrahepatic cholestasis [101,102]. Lp-X can be mistaken for LDL on routine clinical tests [101] because it is similar in lipid composition, with more heterogeneity in APOE content. Its atherogenic potential is not yet determined; however, in patients with elevated Lp-X palmar xanthomas appear, especially in patients with cholestasis, that regress over months after improvement of hypercholesterolemia [103].

### 6.3. Lipoprotein-Associated Phospholipase A2

Lipoprotein-associated phospholipase A2 (Lp-PLA2) is a platelet activating factor acetylhydrolase, which is found to be involved in atherosclerosis as an established and independent risk biomarker of atherosclerosis-related CVD. It is associated with LDL, HDL and Lp(a) [104], presented schematically in Figure 3.

Its function has been researched more thoroughly—it hydrolyzes oxidized phospholipids that are further formed into lysophospholipids and oxidized non-esterified fatty acids. Lysophospholipids promote the expression of adhesion molecules, stimulate cytokine production and attract macrophages to the arterial intima [100], therefore potentiating atherogenesis. Increased Lp-PLA2 activity further amplifies the risk of CVD mediated by Ox-PL/APOB [98]. A variety of studies support the utility of Lp-PLA2 measurements for estimating and further refining CVD risk in adulthood, especially for coronary artery disease [105,106]. Furthermore, patients on statin therapy after ischemic stroke have shown lower Lp-PLA2 compared with those without it [107], but pharmaceutical inhibition of Lp-PLA2 has failed to demonstrate a significant association with improved prognosis in patients with stable coronary artery disease or after an acute coronary syndrome [105].

In children, similar trends have been observed, where Lp-PLA2 concentrations are elevated in children with hypercholesterolemia and associated with other markers of atherosclerosis, such as cholesterol levels and cIMT, suggesting its potential use in identifying early atherosclerotic changes in children [108]. Additionally, in a study of children with heterozygous familial hypercholesterolemia, the activity of Lp-PLA2 was increased and additionally proven to be reduced after statin therapy [109]. The latter was also shown in a study of children with type 1 diabetes mellitus, where statin therapy lowered LDL and Lp-PLA2 activity [110]. Not only genetic causes but also dietary habits influence Lp-PLA2 activity, as demonstrated by its higher activity in children and adolescents with elevated triglycerides/HDL ratio [111] and with obesity [112,113]. Even more concerningly, children with obesity demonstrate levels of Lp-PLA2 activity in the rank of atherosclerosis and a high thromboembolic risk in adulthood [112]. Lp-PLA2 activity is also higher in children with obstructive sleep apnea syndrome [114] or with Kawasaki disease [115]. The clinical utility of Lp-PLA2 has yet to be determined, since so far it has not shown to surpass other markers, such as LDL and APOB, in apparently healthy paediatric populations [116].

### 6.4. Ceramides

Ceramides are components of all cell membranes, and, in addition to their structural function, they are also involved in immune cell function and inflammation [117]. Increased ceramide concentrations are associated with obesity and the development of insulin resistance, type 2 diabetes mellitus and increased CVD risk [117]. They are believed to be dietary biomarkers and products of lipid metabolism which accumulate in individuals with obesity/dyslipidemia and which disrupt cellular processes, eliciting tissue dysfunction leading to diabetes and heart disease [118].

Ceramides belong to sphingolipids. Among these, sphingosine-1-phosphate and sphingomyelin may play an important role in atherogenesis as they were found in atherosclerotic plaque. Furthermore, the inhibition of the de novo ceramide synthesis reduces the development of atherosclerosis [119]. Additionally, genetic studies have uncovered genetic variants in genes involved in ceramide biosynthesis, suggesting a heritable component in the mechanistic insights of the signalling activities of ceramides in vascular inflammation and apoptosis leading to CVD and insulin resistance risk [120]. Evidence is emerging that certain ceramides also mediate cellular mechanisms, leading to the progression of micro- and macrovascular complications of diabetes mellitus [121].

In the adult population, ceramides have been recognized as significant predictors of cardiovascular death among patients with coronary artery disease or acute coronary syndrome, surpassing currently used biomarkers, and have been independently associated with adverse cardiovascular events [122,123]. In children, similarly, elevated ceramides have been observed in obese patients together with increased insulin resistance parameters associated with non-alcoholic fatty liver disease [124]. A dietary component has been seen through fructose restriction, leading to a trend of decreasing ceramides in children with obesity [125]. Interestingly, quantitative differences in ceramides levels have been observed among obese mothers and their children compared to normal-weight mothers and their children [126]. Ceramides are also increased in children with chronic kidney disease, with lactosylceramides recognized as independent predictors of lower systolic function [127].

A specific disease, known in paediatric age, is Farber disease, which was first described in 1957; it is a rare, fatal, inherited metabolic disorder caused by mutations in the lysosomal acid ceramidase. The enzyme is responsible for ceramide degradation, and consequently ceramides accumulate in most tissues in these patients, leading to a spectrum of joint, cardiac, pulmonary and central nervous system defects and death by the age of 2 in the classical form of the disease. In the milder, although rarer, forms, some can survive into young adulthood [128].

## 7. Conclusions

Lipids are a complex group of metabolites; traditionally, when disbalanced, they are regarded as a cardiovascular risk factor. However, due to their complex nature, much research is still needed for a comprehensive understanding of their role in atherosclerosis, especially in the young. Several new lipid biomarkers are emerging; however, for many, even ones that are not so new, the atherogenic potential is not clear and no clinical recommendations are in place to aid the clinician in using them in everyday clinical practice. Additional research to evaluate and to develop practical recommendations is encouraged.

According to our knowledge, our review is the first to provide a comprehensive foundation for further research in lipid biomarker evaluation in the context of CVD in the paediatric population; however, due to the abundance of lipid molecules, not all could be included, which is the greatest limitation of this review. Our proposed list of lipid biomarkers should be further extended according to potential biomarkers found in the future along with their clinical assessment in cardiovascular risk stratification in children. Furthermore, proposed lipid biomarkers should be tested in a larger paediatric cohort with atherosclerosis assessment.

## Figures and Tables

**Figure 1 ijms-24-02237-f001:**
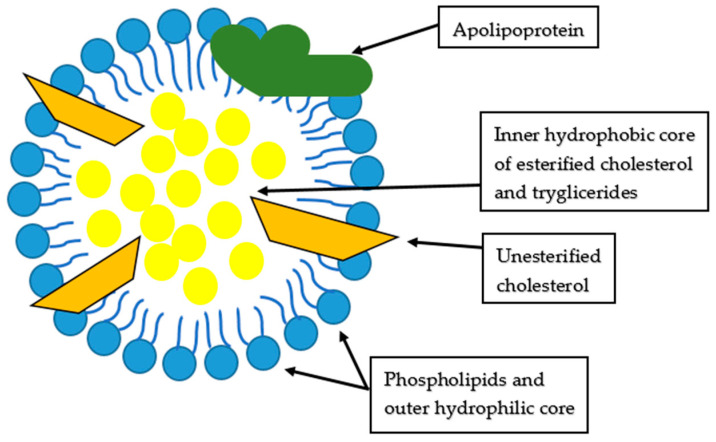
Structure of lipoprotein, modified from Feingold [13].

**Figure 2 ijms-24-02237-f002:**
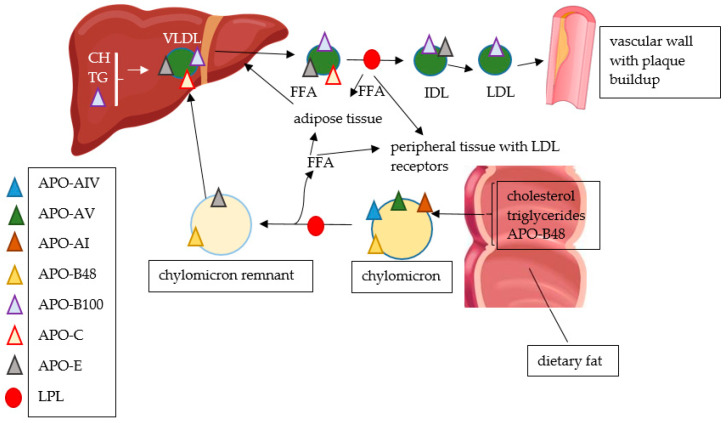
Relationship between dietary fat, basic lipid metabolism and atherogenesis; APO—apolipoprotein, FFA—free fatty acid, LPL—lipoprotein lipase, CH—cholesterol, TG—triglycerides, VLDL—very low-density lipoproteins, IDL—intermediate-density lipoproteins, LDL—low-density lipoproteins.

**Figure 3 ijms-24-02237-f003:**
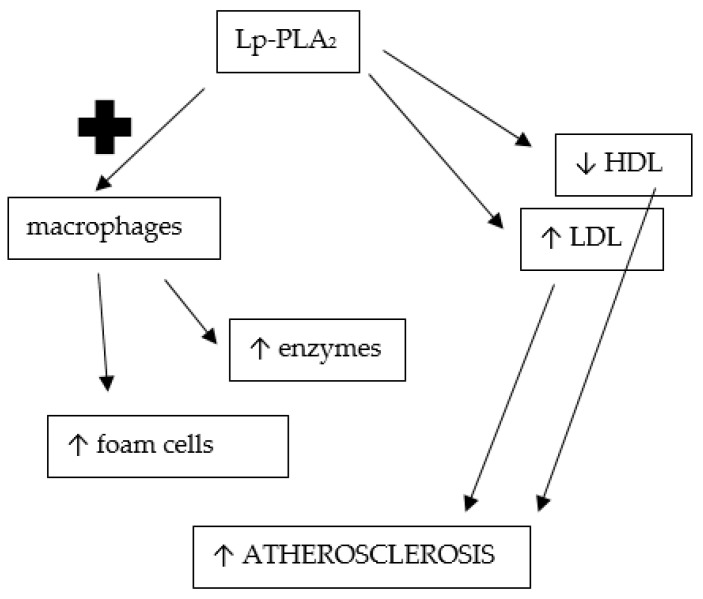
Promotion of atherosclerosis through lipoprotein-associated phosholipase A2 function; Lp-PLA_2_—lipoprotein-associated phosholipase A2, HDL—high-density lipoprotein, LDL—low-density lipoprotein; ↑ — increased; ↓ — decreased.

**Table 1 ijms-24-02237-t001:** Summary of most important previous studies, classified by different criteria and compared to our review.

Review/Original Paper	Research Methods	Applicability of the Results	Comparison to Our Review
Li et al. [4]	Original paper; lipid biomarkers determination in pateints with coronary artery disease	Simultaneous determination of traditional and novel lipid biomarkers showing their association with coronary severity in men	Similarities: simultaneous determination of lipid biomarkers, used for cardiovascular risk stratification
			Differences: research study, adult population
Hinterwirth et al. [5]	Review; lipids classification, lipidomics studies in cohorts of patients	The knowledge of lipid biomarkers lacking, the association between several lipid biomarkers and cardiovascular diseases in adults	Similarities: wide range of biomarkers, review of some disease states
			Differences: applicability to adult population, focus on lipidomic studies
Juhola et al. [12]	Multicenter study; correlation between serum lipid levels in childhood and adulthood	The importance of lipid biomarkers in children	Similarities: the importance of lipid biomarkers in paediatric population
			Differences: our review provides additional novel and research biomarkers, not included previously
Feingold et al. [13]	Review; a comprehensive explanation of lipoprotein and apolipoprotein’s structure with their physiologic function	Physiologic function of several biomarkers	Similarities: comprehensive review
			Differences: inclusion of lipids, other than lipoproteins with studies of cardiovascular risk assessment

**Table 2 ijms-24-02237-t002:** Traditional, newer and research lipid biomarkers; (P)—pro-atherogenic, (A)—anti-atherogenic, (U)—atherogenic function less clear/mixed results.

Traditional Lipid Biomarkers	Newer Lipid Biomarkers	Research Lipid Biomarkers
total cholesterol (P)	lipoprotein(a) (P)	oxidised lipids (lipid peroxidation) (P)
low-density lipoprotein cholesterol (P)	apolipoproteins A: AI (A), AII (U), AIV (U), AV (U)	lipoprotein X (P)
high-density lipoprotein cholesterol (A)	apolipoproteins B: B48 (P), B100 (P)	lipoprotein-associated phospholipase A2 (P)
Triglycerides (P)	apolipoproteins C: CI (P), CII (P), CIII (P)	Ceramides (P)
	apolipoproteins E (P)	

## Data Availability

Not applicable.

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
