# Peer review of "Lipid Biomarkers and Atherosclerosis—Old and New in Cardiovascular Risk in Childhood"

_ijms, 2023, doi:10.3390/ijms24032237_

Round 1
Reviewer 1 Report
In this manuscript, the authors summarize the results from previous studies about lipid biomarkers and atherosclerosis as a factor for cardiovascular risk in childhood.My remarks are as follows:
The motivation part in the abstract and introduction should be improved. The authors’ contributions should be concisely expressed. A comparison with previous similar review papers must be added.
At the end of the “Introduction” section, please add a short description of the paper’s structure.
“2. Traditional lipid biomarkers”, “3. Newer lipid biomarkers” and “4. Research lipid biomarkers” sections are too verbose and should be rewritten. Please, add a discussion section using systematic mapping approach. Here, you should classify the results obtained in previous studies by different criteria (papers’ quality, research methods, applicability of results, etc.).
“5. Conclusions” section should be enhanced. Please, add data about your study limitations. The part concerning further plans also should be extended.
Technical remark:
The label of Figure 1 should be re-formatted. Please, check the specific guidance provided in author instructions.
Author Response
Dear reviewer,
All the answers are provided in the attached file.
Kind regards

Reviewer 2 Report
The review of Močnik and Varda is interesting scientifically. The Authors summarize the current knowledge on lipids as biomarkers of cardiovascular risk in children and indicate a need for the discovery of new, additional lipid biomarkers. The Introduction contains the necessary information regarding the role of lipids in atherosclerosis and the characteristics of this group of compounds. Then, the next three chapters present a state-of-the-art on traditional, newer, and research lipid biomarkers associated with cardiovascular risk in children. In general, the manuscript is well thought out and well organized. The bibliography (most papers from the last decade, up to the 2022 year) is complete. However, the following issues need to be considered prior to considering the manuscript for publication in IJMS.
[Major concern]
- In the Introduction section: There is no explanation for the problem of cardiovascular diseases in paediatric population. Some statistics on the incidence of atherosclerosis in children would be useful here.
- I propose to add somewhere a few sentences about lipid determination methods.
- It would be interesting to compare (e.g., in the figure or in the table) lipid biomarkers of cardiovascular risk in children and adults.
[Minor concern]
- Line 158: Include a number for this section.
- Please keep the format of lipid names consistent in the whole text (e.g., see Table 1 and lines 109, 160-163).
- Line 306: No space after ‘65]’.
With kind regards,
Reviewer
Author Response

(The authors gave the same response as above.)

Reviewer 3 Report
In the present manuscript, the authors have reviewed the lipid biomarkers for the evaluation of cardiovascular risk in the pediatric population. Overall, the manuscript has been written well and provides compiled report for the current understanding of lipid biomarkers in the context of CVD and is of much interest to the readers. My suggestions are as under:
1. Please summarize all the Atherosclerosis lipid biomarkers in the form of a table containing whether their high level or low level is anti-atherogenic or pro-atherogenic in nature with references.
2. It would enhance the quality of the manuscript if the graphical representation of lipid biomarkers-related signaling/ actions could be included in the manuscript.
3. Use either APO-AI (roman) or APO-A1 (numerical), not both.
Author Response

(The authors gave the same response as above.)

Round 2
Reviewer 1 Report
The quality of ijms-2153635-v2 “Lipid biomarkers and atherosclerosis - old and new in cardiovascular risk in childhood” has been improved substantially, especially by adding a new “2. Methods and previous studies” section containing data mapping and analysis of results obtained in previous similar studies.
In my opinion, the manuscript meets the requirements of International Journal of Molecular Sciences.
My recommendation is “Accept as is”.
Reviewer 3 Report
No further comments